# Plasma CXCL8 and MCP-1 as surrogate plasma biomarkers of latent tuberculosis infection among household contacts–A cross-sectional study

Sivaprakasam T. Selvavinayagam[1ᐧ], Bijulal Aswathy[2ᐧ], Yean K. Yong[3ᐧ], Asha Frederick[4], Lakshmi Murali[4], Vasudevan Kalaivani[1], Sree J. Karishma[2], Manivannan Rajeshkumar[1], Adukkadukkam Anusree[5], Meganathan Kannan[5], Natarajan Gopalan[6], Ramachandran Vignesh[7], Amudhan Murugesan[8], Hong Yien Tan[3], Ying Zhang[3], Samudi Chandramathi[9], Munusamy Ponnan Sivasankaran[10], Pachamuthu Balakrishnan[11], Sakthivel Govindaraj[12], Siddappa N. Byrareddy[13], Vijayakumar Velu[12], Marie Larsson[14], Esaki M. Shankar[2]*, Sivadoss Raju[1]*

**1** State Public Health Laboratory, Directorate of Public Health and Preventive Medicine, DMS Campus, Teynampet, Chennai, Tamil Nadu, India, **2** Department of Biotechnology, Infection and Inflammation, Central University of Tamil Nadu, Thiruvarur, India, **3** Laboratory Centre, Xiamen University Malaysia, Sepang, Selangor, Malaysia, **4** National Tuberculosis Elimination Programme, Chennai, Tamil Nadu, India, **5** Department of Life Sciences, Blood and Vascular Biology, Central University of Tamil Nadu, Thiruvarur, India, **6** Department of Epidemiology and Public Health, Central University of Tamil Nadu, Thiruvarur, India, **7** Pre-clinical Department, Royal College of Medicine, Universiti Kuala Lumpur, Ipoh, Malaysia, **8** Department of Microbiology, The Government Theni Medical College and Hospital, Theni, India, **9** Department of Medical Microbiology, University of Malaya, Kuala Lumpur, Malaysia, **10** Seattle Children's Research Institute, Seattle, WA, United States of America, **11** Department of Microbiology, Saveetha Institute of Medical and Technical Sciences (SIMATS), Centre for Infectious Diseases, Velappanchavadi, Chennai, India, **12** Department of Pathology and Laboratory Medicine, Division of Microbiology and Immunology, Emory University School of Medicine, Emory National Primate Research Center, Emory Vaccine Center, Atlanta, GA, United States of America, **13** Department of Pharmacology and Experimental Neuroscience, University of Nebraska Medical Center, Omaha, Nebraska, United States of America, **14** Department of Biomedicine and Clinical Sciences, Linkoping University, Linköping, Sweden

ᐧ These authors contributed equally to this work.
* dphpmlab@gmail.com (SR); shankarem@cutn.ac.in (EMS)

**Data Availability Statement:** All relevant data are within the paper and its Supporting information files.

## Abstract

Early detection of latent tuberculosis infection (LTBI) is critical to TB elimination in the current WHO vision of *End Tuberculosis Strategy*. The study investigates whether detecting plasma cytokines could aid in diagnosing LTBI across household contacts (HHCs) positive for IGRA, HHCs negative for IGRA, and healthy controls. The plasma cytokines were measured using a commercial Bio-Plex Pro Human Cytokine 17-plex assay. Increased plasma CXCL8 and decreased MCP-1, TNF-α, and IFN-γ were associated with LTBI. Regression analysis showed that a combination of CXCL8 and MCP-1 increased the risk of LTBI among HHCs to 14-fold. Our study suggests that CXCL-8 and MCP-1 could serve as the surrogate biomarkers of LTBI, particularly in resource-limited settings. Further laboratory investigations are warranted before extrapolating CXCL8 and MCP-1 for their usefulness as surrogate biomarkers of LTBI in resource-limited settings.

**Funding:** S. T. S. and S. R. are funded by the National Health Mission, Tamil Nadu (680/NGS/NHMTNMSC/ENGG/2021) for the Directorate of Public Health and Preventive Medicine. E.M.S. is funded by the Department of Science and Technology-Science and Engineering Research Board, Government of India (CRG/2019/006096). This work is also supported by grants through AI52731, the Swedish Research Council, the Swedish, Physicians against AIDS Research Foundation, the Swedish International Development Cooperation Agency, SIDA SARC, VINNMER for Vinnova, Linköping University Hospital Research Fund, CALF, and the Swedish Society of Medicine (to ML). H. Y. T. is supported by Xiamen University Research Funding (XMUMRF/2020-C5/ITCM/0003 and VV is supported by: The NIH Office of Research Infrastructure Programs (P51 OD011132 to ENPRC), and Emory CFAR (P30 AI050409). The funders had no role in study design, data collection and analysis, decision to publish, or preparation of the manuscript.

**Competing interests:** The authors have declared that no competing interests exist.

## Introduction

Tuberculosis (TB) is one of the most devastating infectious diseases, resulting in ~1.6 million deaths in 2021. Reports suggest that one-fourth of the global population was infected with *Mycobacterium tuberculosis* (*M. tuberculosis* or MTB) in 2021 [1]. Latent TB infection (LTBI) results in persistent immune responses to MTB antigens without any gross evidence of clinical TB [1,2]. Estimates suggest that 5–10% of individuals with underlying LTBI might progress to develop active TB disease [1,3]. According to the World Health Organization's (WHO) *End Tuberculosis Strategy*, the early detection of LTBI, especially in endemic areas, is key to global TB elimination [4,5]. Hence, an improved method to detect LTBI is urgently required, [4,6] especially in resource-limited settings.

TB diagnostics suffers from lack of a gold standard test [7]. Until the advent of the interferon-gamma-release assay (IGRA), the tuberculin skin test (TST) was the only tool available [8,9]. TST uses a purified protein derivative (PPD) antigen [9]. It endures poor sensitivity for use in immune-compromised individuals and poor specificity due to several confounding factors [10,11]. IGRA appears more specific than the TST for detecting LTBI [5,12]. In addition, an individual's immune status could influence IGRA results as suggested by others [8,13]. However, IGRA and TST cannot distinguish between active TB and LTBI [2,11,14]. Hence, to address these limitations, identifying an alternative molecular biomarker is necessary [5,7].

During MTB infection, macrophages regulate cytokines secreted by T cells [15]. In concert with complex mycobacterial antigens, cytokines mount protective and pathogenic responses [16]. Previous literature has suggested that specific cytokines could aid in detecting LTBI, and highlighted the likely role of IFN-α, TNF-α, MCP-1, MIP-1β, IL-2, CXCL-8, IL-6, and GM-CSF with TB disease progression [5,12]. Here, we investigated if cytokines could serve as surrogate biomarker(s) of LTBI among household contacts (HHCs) of individuals with active TB disease.

## Materials and methods

### Ethical approval

This study was performed within the ethical standards of the Declaration of Helsinki. The study procedures and/or protocols were reviewed and approved by the Human Ethics Committee of the Directorate of Public Health and Preventive Medicine Ethical Committee (DPH, IEC 15-10-2022), Chennai, India. All participants provided written informed consent to participate in the study. The start date of the participant recruitment was on 13th February 2023 and the end date was 17th February 2023. An algorithmic layout of the study is presented in Fig 1.

### Clinical samples and study design

The cross-sectional study recruited 76 volunteers by a random sampling method (from a total of 172 individuals) who were further divided into three cohorts: Household contacts/IGRA+ve (HHC/IGRA+ve) (n = 26 from 58 positive individuals), household contacts/IGRA-ve (HHC/IGRA-ve) (n = 25 from 52 negative individuals) and healthy controls (HCs) (n = 25 from 44 individuals tested negative for LTBI). HCs were defined as having had no contact with active TB cases and were negative for IGRA.

### Laboratory analytes

Blood glucose levels, liver analytes, renal parameters and CRP levels were measured using standard routine laboratory protocols. Total bilirubin, SGPT, SGOT, ALP, albumin, globulin and

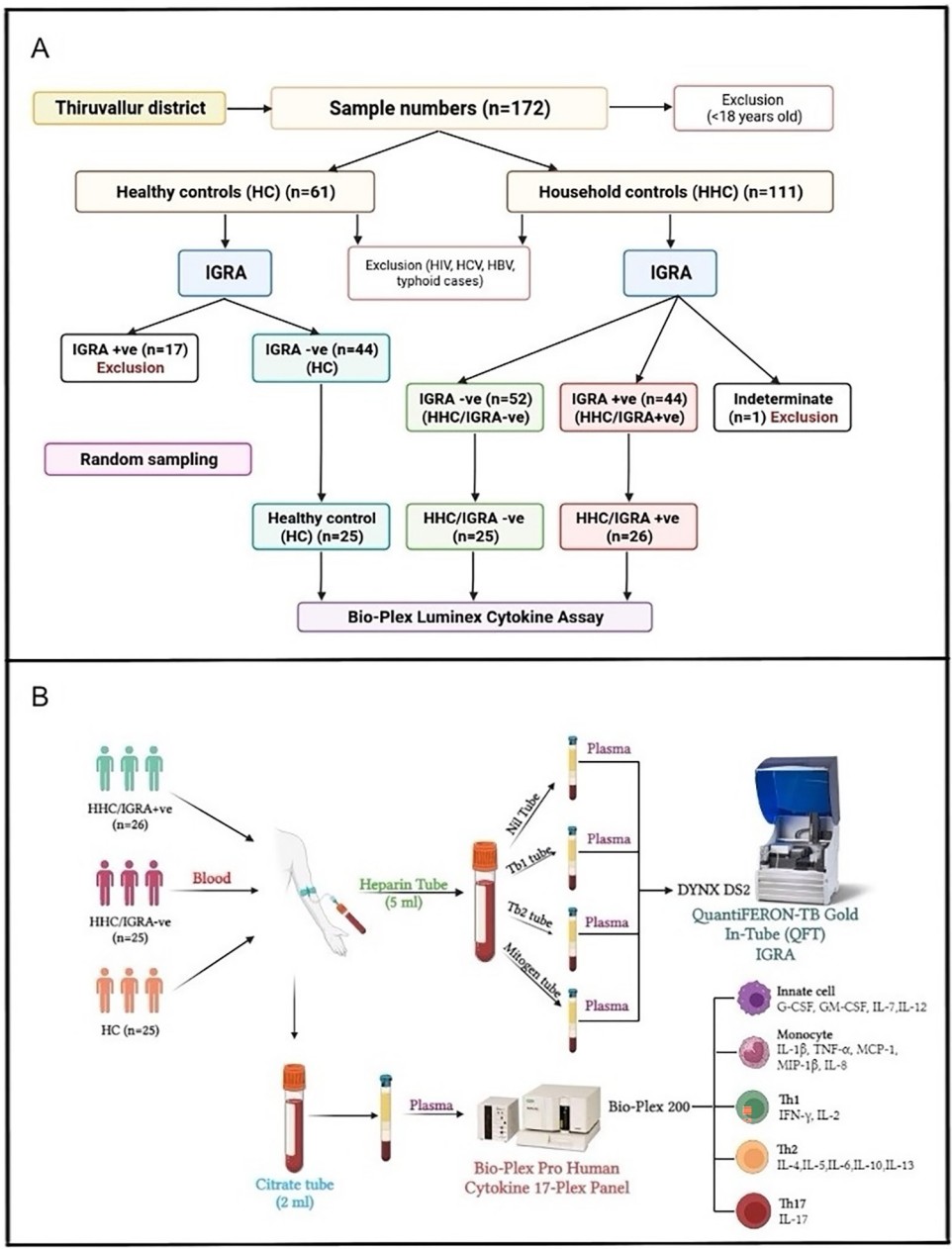

**Fig 1. Algorithm of sampling from the study cohort and outline of the investigation. A)** The flow-chart outlining the cross-sectional study sampling for the LTBI cohort. The study recruited 76 volunteers by a random sampling method, the exclusion criteria's as well as the IGRA reports (from a total of 172 individuals) who were further divided into three cohorts: Household contacts/IGRA+ve (HHC/IGRA+ve) (n = 26 from 58 positive individuals), household contacts/IGRA-ve (HHC/IGRA-ve) (n = 25 from 52 negative individuals) and healthy controls (HCs) (n = 25 from 44 individuals tested negative for LTBI). HCs were defined as having had no contact with active TB cases and were negative for IGRA. **B)** Blood was drawn in Lithium-heparin tubes and Sodium-citrate tubes. Plasma samples in the Lithium-heparin tubes were subjected to IGRA after transferring them to the QFT tubes. Plasma samples in the sodium-citrate tubes were subjected to Bio-Plex Luminex Cytokine assay.

total protein were the liver parameters studied whereas uric acid, blood urea nitrogen (BUN), urea and creatinine were the renal function analytes examined using an automated Biochemistry Analyzer (Siemens, Germany).

### Measurement of IFN-γ by QuantiFERON-TB Gold In-Tube Assay

Blood samples (lithium heparin (5 ml) and sodium citrate (2 ml) BD Vacutainer tubes) were obtained from 172 individuals who were divided primarily into two cohorts: HCs and HHCs. As per the manufacturer's instructions, all 172 samples were subjected to a commercial QuantiFERON-TB Gold In-Tube Assay (Cat. No.: 622130, Qiagen, USA). Briefly, heparinized plasma was aliquoted into four QFT-Plus blood tubes, viz., nil tube, TB antigen tube 1 (TB1) possessing ESAT-6 and CFP-10 peptides to activate CD4+ T cells, TB antigen tube 2 (TB2) that includes ESAT-6 and CFP-10 peptides to activate CD8+ T cells and mitogen tube as a positive control. The aliquoted tubes were mixed ~10 times before 16–24 hours of incubation for 15 min at 3000 rpm. Subsequently, the separated plasma was subjected to an in-built ELISA to detect IFN-γ. The results were calculated using QFT Plus analysis software by analyzing the IFN-γ level of the post-reaction supernatant. The results were interpreted as positive, negative, or indeterminate [4,9].

### Cytokine assay

To quantify the levels of various plasma cytokines, we used a commercial Bio-plex Pro Human Cytokine 17-plex assay (Bio-Rad Laboratories, Hercules, CA) that uses sodium-citrated plasma without TB antigen stimulation. The kit measures the following cytokines: IL-1β, IL-2, IL-3, IL-4, IL-5, IL-6, IL-7, CXCL8, IL-10, IL-12, IL-13, IL-17A, G-CSF, GM-CSF, IFN-γ, MCP-1, MIP-1 and TNF-α) as per the manufacturer's instructions. The data were analyzed using the Bio-plex Manager Software Ver.6.1.

### Statistical analysis

The comparison of the group differences was made using the Mann-Whitney test. The Kruskal-Wallis test was used for testing statistical significance with the median values. The Chi-Square test was used to compare the categorical variables. Receiver operating characteristic (ROC) curves were drawn to define the diagnostic performance of the biomarker. The best cut-off value (sum of sensitivity and specificity divided by 100) was chosen to maximize Youden's index and the AUC. GraphPad Prism 6.0 (GraphPad Software, San Diego, CA), SPSS version 20.0, and Microsoft Excel 2019 were used for the statistical evaluation. The level of significance was $p < 0.05$.

## Results

### Study characteristics

The group characteristics revealed considerable differences among the male and female participants, particularly among males. All the participants were adult subjects aged >18 years. The clinico-demographic data obtained from the participants are presented in Table 1. Of all the parameters studied, we observed a trend of low median level of SGOT, SGPT, and urea among HHCs as compared to HCs. The study comprised 13 individuals in the HHC/IGRA+ve group (n = 26), 12 individuals each in the HHC/IGRA-ve group (n = 25) and the HC group (n = 25) with underlying conditions. The socioeconomic characteristics of the cohort are presented in S1 Fig.

### An increase of CXCL8 and a decrease of MCP-1, TNF-α, and IFN-γ were associated with LTBI

Seven of the 17 cytokines measured using the Bio-Plex Luminex cytokine assay remained undetectable, viz., IL-2, IL-4, IL-5, IL-7, IL-12, G-CSF, and GM-CSF. Hence, we incorporated only the remaining ten cytokines in the analysis. Of the monocyte-derived cytokines, CXCL8,

**Table 1. Clinico-demographic characteristics of the study cohort.**

| Characteristics | HC | HHC/IGRA+ve | HHC/IGRA-ve | *p* value |
|---|---|---|---|---|
| **Number** | 25 | 26 | 25 | |
| **Age,** years, *median (IQR)* | 36 (29–47) | 39.5 (28–54.8) | 40 (33–47) | 0.492 |
| **Gender,** male, *n (%)* | 14 (56%) | 9 (34.6%) | 7 (28%) | 0.044* |
| **BMI,** *median (IQR)* | 25.4 (22.9–28.4) | 25.8 (23–30.23) | 24.7 (21.9–28.4) | 0.748 |
| **BCG vaccination,** *n (%)* | 21 (84%) | 21 (80.8%) | 20 (80%) | 0.379 |
| **Residential area,** urban, *n (%)* | 13 (52%) | 13 (50%) | 13 (52%) | 0.986 |
| **Bilirubin,** (mg/dl), *median (IQR)* | 0.7 (0.6–0.85) | 0.7 (0.6–0.83) | 0.6 (0.45–0.9) | 0.493 |
| **SGOT,** (U/L), *median (IQR)* | 25 (16–32) | 21.5 (18–26.5) | 20 (17–25.5) | 0.078[†] |
| **SGPT,** (U/L), *median (IQR)* | 25 (16–43.5) | 19.5 (16–26) | 19 (14–22.5) | 0.075[†] |
| **ALP,** (U/L), *median (IQR)* | 69 (51–74) | 69 (54.8–90.3) | 58 (49–73) | 0.109 |
| **Creatinine,** (mg/dl), *median (IQR)* | 0.8 (0.8–1.0) | 0.8 (0.7–0.9) | 0.8 (0.7–0.9) | 0.152 |
| **Urea,** (mg/dl), *median (IQR)* | 23 (20–34) | 22.5 (16.7–25.8) | 20 (17–23.5) | 0.052[†] |
| **Neutrophil count** $(10x^3/\mu L)$, *median (IQR)* | 4.57 (4.07–5.31) | 4.4 (3.4–6.5) | 4.2 (3.7–5.53) | 0.693 |
| **%,** *median (IQR)* | 59.7 (52–63.5) | 60.6 (43.4–66.3) | 55.1 (50.5–62.9) | 0.954 |
| **Lymphocyte count** $(10x^3/\mu L)$, *median (IQR)* | 2.44 (2.07–2.81) | 2.62 (2.07–3.32) | 2.2 (1.8–2.86) | 0.263 |
| **%,** *median (IQR)* | 29.6 (27.3–53.6) | 31.2 (25.1–39.6) | 31.1 (24.6–35.9) | 0.925 |
| **Monocyte count** $(10x^3/\mu L)$, *median (IQR)* | 0.52 (0.43–0.65) | 0.57 (0.43–0.64) | 0.51 (0.41–0.7) | 0.974 |
| **%,** *median (IQR)* | 6.6 (5.6–7.8) | 6.85 (5.15–7.65) | 6.6 (5.8–9.1) | 0.828 |
| **Eosinophil count** $(10x^3/\mu L)$, *median (IQR)* | 0.19 (0.11–0.27) | 0.25 (0.18–0.48) | 0.2 (0.4–0.39) | 0.166 |
| **%,** *median (IQR)* | 2.3 (1.35–4.25) | 3.55 (1.93–6.7) | 2.9 (1.65–5.05) | 0.189 |
| **Basophil count** $(10x^3/\mu L)$, *median (IQR)* | 0.04 (0.03–0.06) | 0.05 (0.04–0.06) | 0.04 (0.03–0.06) | 0.384 |
| **%,** *median (IQR)* | 0.6 (0.45–0.8) | 0.6 (0.5–0.9) | 0.6 (0.5–0.75) | 0.719 |
| **Medical conditions,** *n (%)* | | | | |
| Sickness past 14-days | 6 (24%) | 4 (15.4%) | 4 (16%) | 0.479 |
| Diabetes mellitus | 5 (20%) | 8 (30.8%) | 6 (24%) | 0.753 |
| Hypertension | 6 (24%) | 4 (15.4%) | 4 (16%) | 0.479 |
| COPD/asthma | 2 (8%) | 2 (7.7%) | 3 (12%) | 0.632 |
| Hemodialysis | 2 (8%) | 0 (0%) | 2 (8%) | 1.000 |
| History of COVID-19 | 6 (24%) | 4 (15.4%) | 4 (16%) | 0.450 |
| Smoking | 2 (8%) | 1 (3.85%) | 0 (0%) | 0.153 |
| Alcohol | 4 (16%) | 2 (7.7%) | 0 (0%) | 0.141 |

All categorical variable reported as numbers (n) and percentages (%), and continuous variables reported as median, IQR. HC, healthy control; HHC, household contact, IGRA, interferon gamma release assay; SGOT, serum glutamic oxaloacetic transaminase; SGPT, serum glutamic-pyruvic transaminase; ALP, alkaline phosphatase. All continuous variables were compared using the Kruskal-Wallis test, whilst all the categorical variables were compared using Chi-Square test. *Represent $p<0.05$, [†]having a trend of significance.

MCP-1, and TNF-α showed significance. MIP-1 and IL-1 did not show any significant difference. Similarly, IFN-γ showed a significant difference among the T-cell-derived cytokines. IL-6, IL-10, IL-17A, and IL-13 did not reveal any marked differences. CXCL8 levels in HHC/IGRA+ve and HHC/IGRA-ve were higher than HCs ($p<0.05$). MCP-1 levels in HHC/IGRA+ve and HHC/IGRA-ve were lower than the HCs ($p<0.01$). TNF-α levels in the HCC/IGRA+ve and HHC/IGRA-ve were lower than HCs ($p<0.001$). IFN-γ concentrations in HHC/IGRA+ve and HCC/IGRA-ve were lower than HCs (Fig 2).

## CXCL8 and MCP-1 predicted the risk of the development of LTBI

To assess the suitability of CXCL8, MCP-1, TNF-α, and IFN-γ as surrogate biomarkers for the detection of LTBI, the receiver-operating characteristics (ROC) analysis was performed between HHC/IGRA+ve and HCs. Our analysis showed that CXCL8 and MCP-1 could predict LTBI with the area under the curve (AUC) of 0.6885; $p = 0.0210$ and 0.7392; $p = 0.0034$,

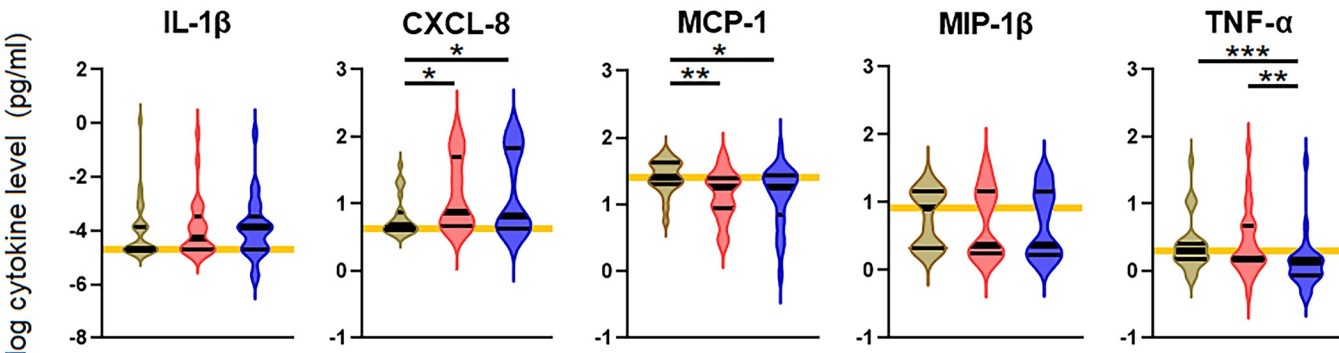

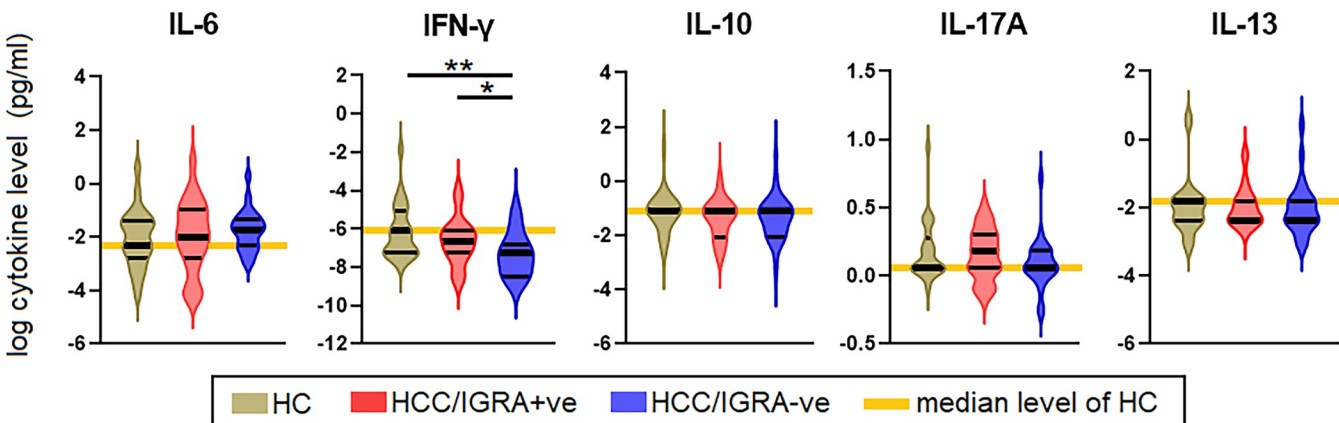

**Fig 2. Comparison of the levels of cytokines among HC, HHC/IGRA+ve and HHC/IGRA-ve individuals. A)** Monocyte-derived cytokines **B)** T cell-derived cytokines. IL, interleukin; MCP-1, monocyte chemoattractant protein-1; MIP-1β, macrophage inflammatory protein-1 beta; TNF-α, tumor necrosis factor alpha; IFN-γ, interferon gamma. *, ** and *** represent $p<0.05$, $<0.01$, $<0.001$, respectively.

respectively. From the ROC analysis, we determined the cut-off for CXCL8 and MCP-1. The cut-off value for CXCL8 was >6.2 pg/ml, while for MCP-1 it was <22.56 pg/ml. Though the AUC for CXCL8 and MCP-1 were seemingly low when we combined both CXCL8 and MCP-1, the AUC was 0.9393; $p<0.0001$ (Fig 3A).

A binary regression analysis was performed to examine further the relationship between CXCL8 and MCP-1 and the likelihood of developing LTBI. We found that a combination of both CXCL8 (>6.2 pg/ml) and MCP-1 (<22.56 pg/ml) was associated with increased risk of LTBI by 14-fold (95% CI = 2.82–69.6); $p = 0.001$. Considering the levels of cytokines, which may change as age increases, we performed a multivariate binary regression analysis adjusting for age. The results showed that the CXCL8 (>6.2 pg/ml) and MCP-1 (<22.56 pg/ml) remained significant even with the change of age, where the odds was 15.22 (95% CI = 2.9-79-8); $p<0.0001$ (Fig 3B).

## Discussion

LTBI has become a severe public health concern, not only because of the number of people that have already become latently infected with MTB but because of the risk of reactivation.

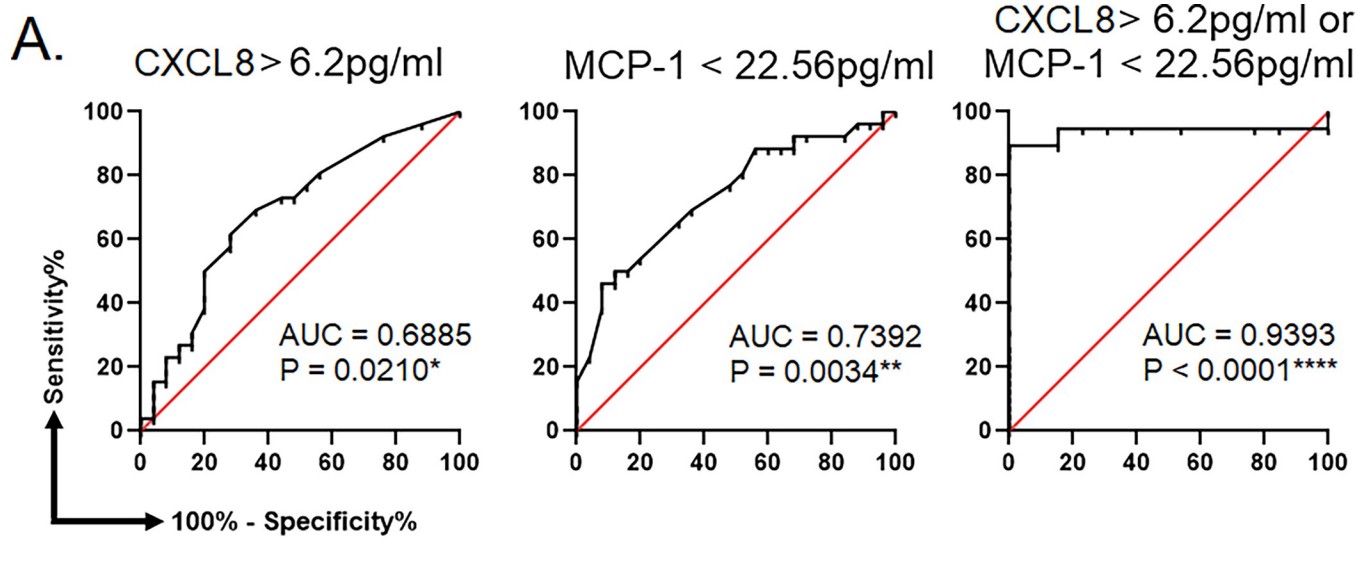

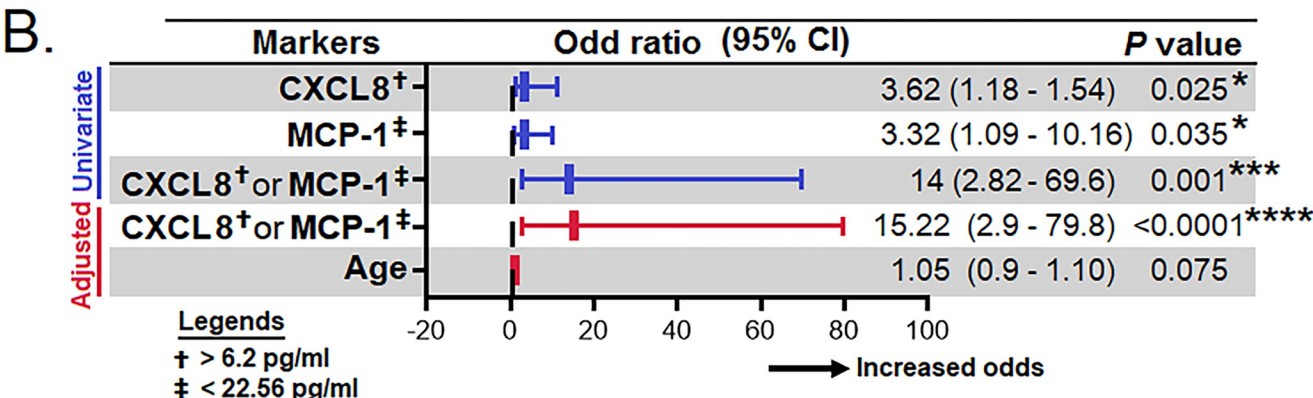

**Fig 3. Efficacy of surrogate biomarkers in predicting LTBI. A)** Receiver operating characteristic curves for the prediction of LTBI by using IL-8, MCP-1 and, either IL-8 or MCP-1. **B)** Association of IL-8, MCP-1 and, either IL-8 or MCP-1 with the risk of LTBI. These analyses were done between HC (true negative) and HCC/IGRA+ve (true LTBI). AUC, area under curve; CI, confident interval; IL, interleukin; MCP-1, monocyte chemoattractant protein-1. *, **, *** and, **** represent $p<0.05$, $<0.01$, $<0.001$ and $<0.0001$, respectively.

>80% of the active TB cases reported in the US are due to reactivation, which can be prevented if effective screening tools are available to initiate timely treatment [17]. An ideal biomarker needs to identify individuals with LTBI without needing *ex vivo* antigenic stimulation. This is because stimulation with MTB antigens such as PPD, CFP-10, and ESAT-6 would either have a strong cross-reactivity with BCG-vaccinated individuals or Normal (Web) with someone with a history of MTB and hence skewing towards false-positive results. Furthermore, *ex vivo* stimulation usually does not work well in immunosuppressed or HIV-infected individuals [14].

The study examined a panel of cytokines for their potential to predict LTBI without ex vivo antigen stimulation. Two potential surrogate markers, CXCL8 and MCP-1, with a combined overall efficacy of ~90% were identified. As these markers could distinguish LTBI from HC without *ex vivo* antigen stimulation, using these markers may be cost-effective and less laborious, especially in resource-limited settings [18]. An important mechanism in TB pathogenesis is granuloma formation, a cellular response in which macrophages accumulate locally in the

lungs along with other leukocytes to "wall off" MTB from disseminating to host tissues [19]. This is a complex process where multiple chemokines, especially CXCL8 and MCP-1 (or CCL-2), regulate leukocyte influx to the TB-infected sites [20,21]. However, MTB is a highly successful pathogen that has evolved several strategies to evade host immunosurveillance to ensure its persistence in the host. The early secreted antigenic target 6 kDa (ESAT-6) is a virulence factor in MTB that significantly influences disease pathogenesis. ESAT-6 inhibits functional antigen-presenting cell responses by reducing IL-12 production by macrophages [22] via their lysis [23,24], destabilizing phagolysosome to allow MTB to escape phagosome, [25] and promoting their intracellular dissemination [24,26] Dissemination of ESAT-6 within macrophage cytosol could block the interaction between MyD88 and IRAK4 to prevent NF-κB activation from causing the attrition of IL-12, IL-6, IFN-γ, and TNF-α [27,28].

Studies have shown that exposure to THP-1 with MTB and serum from active TB patients was associated with elevated CXCL8 and MCP-1 levels [29]. However, we observed that individuals with LTBI had increased plasma CXCL8 but decreased MCP-1 levels. Such difference suggests that the MTB in HHCs might have altered the host immune responses to establish latency. We observed that both IFN-γ and TNF-α were lowered in LTBI individuals as compared to HCs. Others have shown that mycobacterial antigen-induced CXCL8 levels declined with preventive treatment, offering hints for evaluating newer prognostic biomarkers to assess performance [29]. The increase of CXCL8 levels among both IGRA positive and negative groups than HCs in the unstimulated plasma suggests that CXCL8 might be secreted due to an ongoing MTB infection. Further, the binary regression analysis showed that the combination of CXCL8 and MCP-1 was associated with an increased risk of LTBI by 14-fold. Also, HHC/IGRA-ve individuals, albeit IGRA was negative, their cytokines profile was akin to HHC/IGRA+ve individuals indicating that they may have been in contact with MTB or LTBI.

Notwithstanding our investigation underpins the potential role of CXCL8 and MCP-1 in LTBI, large-scale epidemiological studies involving diverse cohorts for determining the usefulness of CXCL8 and MCP-1 as biomarkers of LTBI are warranted. Our study has some other minor limitations; the first and foremost being the lack of a proper reference standard. Diagnostic tests often suffer from poor performance given that individuals with microbiologically-confirmed TB may still remain negative by the TST, QFT, or T spot tests for ambiguous reasons. Furthermore, the host immune responses that is initially proinflammatory (cell-mediated), eventually shifts towards anergy in TB disease, and therefore it is expected that any diagnostic test relying on host immune responses will vacillate depending on immune dynamics. Besides, there are several other confounding factors especially among asymptomatic individuals (viz., co-infections, smoking, nutrition, socioeconomic factors, helminthic infections, etc.) that could have implications in determining the immune variance. Hence, given the heterogeneous host immune attributes, a larger sample size would have been encouraging in the current investigation.

In conclusion, our current study recommends prospective clinical evaluation of the biomarkers identified herein given that others have previously linked the association of CXCL8 and MCP-1 levels with pulmonary TB [30]. We suggest that more laboratory investigations need to be undertaken in different laboratory settings to evaluate the diagnostic relevance and usefulness of CXCL8, MCP-1, TNF-α, and IFN-γ as surrogate biomarkers of LTBI in resource-limited settings.

## Conclusion

In conclusion, the results identified CXCL8 and MCP-1 that could help identify LTBI in the population. The combination of both CXCL8 and MCP-1 increased the risk of LTBI among

HHCs 14-fold. Together, it could be construed that CXCL8 and MCP-1 could serve as surrogate biomarkers of LTB disease. The role of CXCL8 and MCP-1 as surrogate biomarkers warrant further validation for the possible detection of individuals with LTBI in the general population.

## Supporting information

**S1 Fig. Socioeconomic characteristics of cohort participants. A)** Education level of participants. **B)** Family annual income of the participants. **C)** Relationship of the active TB index case with the participants. **D)** Proximity of the participants with the active TB index case. **E)** Contact duration of the participants with the active TB index case. HC, healthy control; HHC, household contact; IGRA, interferon gamma releasing assay.
(TIF)

## Author Contributions

**Conceptualization:** Sivaprakasam T. Selvavinayagam, Siddappa N. Byrareddy, Vijayakumar Velu, Marie Larsson, Esaki M. Shankar, Sivadoss Raju.

**Data curation:** Sivaprakasam T. Selvavinayagam, Yean K. Yong, Lakshmi Murali, Vasudevan Kalaivani, Adukkadukkam Anusree, Natarajan Gopalan, Amudhan Murugesan, Sakthivel Govindaraj, Esaki M. Shankar.

**Formal analysis:** Bijulal Aswathy, Yean K. Yong, Meganathan Kannan, Natarajan Gopalan, Ramachandran Vignesh, Hong Yien Tan, Sakthivel Govindaraj, Esaki M. Shankar, Sivadoss Raju.

**Funding acquisition:** Sivadoss Raju.

**Investigation:** Sivaprakasam T. Selvavinayagam, Bijulal Aswathy, Asha Frederick, Lakshmi Murali, Vasudevan Kalaivani, Sree J. Karishma, Manivannan Rajeshkumar, Adukkadukkam Anusree, Meganathan Kannan, Sivadoss Raju.

**Methodology:** Bijulal Aswathy, Vasudevan Kalaivani, Sree J. Karishma, Manivannan Rajeshkumar, Adukkadukkam Anusree, Meganathan Kannan, Amudhan Murugesan, Sivadoss Raju.

**Project administration:** Sivaprakasam T. Selvavinayagam, Lakshmi Murali, Vasudevan Kalaivani, Manivannan Rajeshkumar, Natarajan Gopalan, Amudhan Murugesan, Samudi Chandramathi, Siddappa N. Byrareddy, Vijayakumar Velu, Esaki M. Shankar, Sivadoss Raju.

**Resources:** Sivaprakasam T. Selvavinayagam, Yean K. Yong, Asha Frederick, Vasudevan Kalaivani, Manivannan Rajeshkumar, Adukkadukkam Anusree, Natarajan Gopalan, Ramachandran Vignesh, Amudhan Murugesan, Hong Yien Tan, Ying Zhang, Samudi Chandramathi, Munusamy Ponnan Sivasankaran, Pachamuthu Balakrishnan, Siddappa N. Byrareddy, Vijayakumar Velu, Marie Larsson.

**Software:** Yean K. Yong, Asha Frederick, Natarajan Gopalan, Hong Yien Tan, Munusamy Ponnan Sivasankaran, Pachamuthu Balakrishnan, Vijayakumar Velu, Marie Larsson.

**Supervision:** Sivaprakasam T. Selvavinayagam, Meganathan Kannan, Vijayakumar Velu, Esaki M. Shankar, Sivadoss Raju.

**Validation:** Sivaprakasam T. Selvavinayagam, Yean K. Yong, Asha Frederick, Lakshmi Murali, Meganathan Kannan, Natarajan Gopalan, Ramachandran Vignesh, Amudhan Murugesan,

Hong Yien Tan, Ying Zhang, Samudi Chandramathi, Munusamy Ponnan Sivasankaran, Pachamuthu Balakrishnan, Sakthivel Govindaraj, Siddappa N. Byrareddy, Vijayakumar Velu, Marie Larsson, Esaki M. Shankar, Sivadoss Raju.

**Visualization:** Sivaprakasam T. Selvavinayagam, Yean K. Yong, Lakshmi Murali, Manivannan Rajeshkumar, Adukkadukkam Anusree, Meganathan Kannan, Ying Zhang, Pachamuthu Balakrishnan, Sakthivel Govindaraj, Vijayakumar Velu, Marie Larsson.

**Writing – original draft:** Sivaprakasam T. Selvavinayagam, Bijulal Aswathy, Yean K. Yong, Ramachandran Vignesh, Hong Yien Tan, Ying Zhang, Samudi Chandramathi, Munusamy Ponnan Sivasankaran, Sakthivel Govindaraj, Siddappa N. Byrareddy, Vijayakumar Velu, Esaki M. Shankar, Sivadoss Raju.

**Writing – review & editing:** Sivaprakasam T. Selvavinayagam, Bijulal Aswathy, Yean K. Yong, Asha Frederick, Lakshmi Murali, Vasudevan Kalaivani, Sree J. Karishma, Meganathan Kannan, Natarajan Gopalan, Ramachandran Vignesh, Amudhan Murugesan, Hong Yien Tan, Ying Zhang, Samudi Chandramathi, Munusamy Ponnan Sivasankaran, Pachamuthu Balakrishnan, Sakthivel Govindaraj, Siddappa N. Byrareddy, Vijayakumar Velu, Marie Larsson, Esaki M. Shankar, Sivadoss Raju.

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
