## [Decision Letter · Decision Letter 0]

25 Sep 2023

PGPH-D-23-01425

Plasma CXCL8 and MCP-1 as biomarkers of latent tuberculosis infection

Dear Dr. Selvavinayagam,

Thank you for submitting your manuscript to PLOS Global Public Health. After careful consideration, we feel that it has merit but does not fully meet PLOS Global Public Health’s publication criteria as it currently stands. Therefore, we invite you to submit a revised version of the manuscript that addresses the points raised during the review process.

We look forward to receiving your revised manuscript.

Kind regards,

Wilber Sabiiti

Academic Editor

Journal Requirements:

Additional Editor Comments (if provided):

Dear Dr Selvavinayagam and Team

Thank you for submitting your paper to PLOS Global Public Health. After careful consideration of the peer reviews and my own assessment, I recommend a major revision. Please take time to address all comments from reviewers. In addition to addressing the peer review comments, please address the following comments too:

Can you please clarify why you decided to perform random sampling again on the participants to select out those for Bio-Plex assay? . The information in the flow diagram in figure 1A is inconsistent with the body text i.e. you mention 26 IGRA positives yet in the flow diagram, it's all the 58 IGRA positive participants.

Table 1: Participants included in all groups had underlying medical conditions. Although this was not statistically different, this could hugely confound the results. You only highlighted on page 4 in section 3.1 about the 13 IGRA +ve individuals with underlying medical conditions but in the table 1 even the healthy had underlying conditions. The participants were enrolled within a period of one week, it is not clear how and when you performed the enrolments for healthy controls – it may not matter but these individuals might not be as healthy as they indicated. They can no longer be considered healthy if they have any other underlying conditions.

Sincerely

Wilber Sabiiti, PhD

Academic Editor

The word 'postulate' in abstract conclusion 'We postulated that CXCL8 and MCP-1 could be the surrogate biomarkers of LTBI, especially in resource-limited settings', does not make the conclusion meaningful. Please consider rephrasing.

Reviewers' comments:

Reviewer's Responses to Questions

**Comments to the Author**

1. Does this manuscript meet PLOS Global Public Health’s publication criteria? Is the manuscript technically sound, and do the data support the conclusions? The manuscript must describe methodologically and ethically rigorous research with conclusions that are appropriately drawn based on the data presented.

Reviewer #1: Yes

Reviewer #2: Yes

2. Has the statistical analysis been performed appropriately and rigorously?

Reviewer #1: Yes

Reviewer #2: Yes

3. Have the authors made all data underlying the findings in their manuscript fully available (please refer to the Data Availability Statement at the start of the manuscript PDF file)?

Reviewer #1: Yes

Reviewer #2: Yes

4. Is the manuscript presented in an intelligible fashion and written in standard English?

Reviewer #1: No

Reviewer #2: Yes

5. Review Comments to the Author

Reviewer #1: The manuscript addresses an important topic identification of a biomarker to differentiate latent TB

The manuscript requires improvement in English writing style

The title seems incomplete, consider rewriting the title

In the abstract the conclusion looks more of a hypothetical statement I recommend that you rewrite the conclusion.

What was the aim of the study? The aim is not clearly written

The introduction does show the relationship between latent TB and theCXCL8 and MCP-1 and its importance in TB immunity and hence a potential biomarker

The results show a decrease in TNF-α, and IFN-γ. These cytokines have not been mentioned in the introduction and no relationship with the chemokines was mentioned. Furthermore the title does not reflect other cytokines

The methods section does not show analysis of biochemical test however they are included in the results section.

The ROC curve analysis is not clear figure A shows three ROC curve analyses.

Was the analysis between the

LTBI and negative house hold contacts

LTBI and HC

HHC and HC

In your discussion you have this statement, “However, we showed that individuals with LTBI had increased plasma CXCL8 but decreased MCP-1 level” s. What is your deduction of the relationship between these two chemokines and the possible cause of the altered immunity ?

Is the conclusion based on the predictions mad e by the roc analysis?

Reviewer #2: The manuscript by Selvavinayagam et al. presents the results of multiplex assessment of cytokines/chemokines of three cohorts of participants: household tuberculosis contacts with IGRA positive, IGRA negative result, and healthy controls. The assessment demonstrated significant differences in plasma concentration of two cytokines (CXCL8 and MCP-1) which makes them potential biomarkers of latent tuberculosis infection.

The results are interesting as many attempts to identify non-sputum-based biomarkers of tuberculosis infection and disease are being made. It is great to see two more potential biomarkers, however, the problem of this type of studies is that once the biomarkers are identified, they need to be tested on prospective cohorts of individuals to be validated. This is often lacking. I am wondering what the authors of the manuscript would suggest in terms of validation of identified biomarkers and their clinical application.

As for the study design, my concern is how the study cohorts were formed. In some cohorts all participants were included, in others – only half of available participants. Why was it random selection and how can you be sure that there was no selection bias?

Figure 2: not sure if some of these characteristics should be presented in Figure as they seem unrelated to LTBI. Usually the demographic data are presented as a table.

I would also suggest to consider to re-formatting this manuscript as a short research letter rather than a full-text research article.

6. PLOS authors have the option to publish the peer review history of their article (what does this mean?). If published, this will include your full peer review and any attached files.

**Do you want your identity to be public for this peer review?** For information about this choice, including consent withdrawal, please see our Privacy Policy.

Reviewer #1: **Yes: **ESTER ACEN

Reviewer #2: No

---

## [Decision Letter · Decision Letter 1]

1 Nov 2023

Detection of surrogate plasma biomarkers of latent tuberculosis infection among household contacts of individuals positive and negative for the interferon gamma release assay – A cross-sectional study

PGPH-D-23-01425R1

Dear Dr  Shanker

We are pleased to inform you that your manuscript 'Detection of surrogate plasma biomarkers of latent tuberculosis infection among household contacts of individuals positive and negative for the interferon gamma release assay – A cross-sectional study' has been provisionally accepted for publication in PLOS Global Public Health.

Best regards,

Wilber Sabiiti

Academic Editor

Dear Dr Esaki Muthu Shankar

Thank you for submitting your manuscript to PLOS Global Public Health. After careful consideration of reviews from the 4 peer reviewers, I am pleased to accept your manuscript for publication in PLOS Global Public Health. Before your manuscript goes to production, please take time to address any outstanding comments from all reviewers and particularly 3 and 4.

Reviewer Comments (if any, and for reference):

Reviewer's Responses to Questions

**Comments to the Author**

1. If the authors have adequately addressed your comments raised in a previous round of review and you feel that this manuscript is now acceptable for publication, you may indicate that here to bypass the “Comments to the Author” section, enter your conflict of interest statement in the “Confidential to Editor” section, and submit your "Accept" recommendation.

Reviewer #1: All comments have been addressed

Reviewer #2: All comments have been addressed

Reviewer #3: (No Response)

Reviewer #4: (No Response)

2. Does this manuscript meet PLOS Global Public Health’s publication criteria? Is the manuscript technically sound, and do the data support the conclusions? The manuscript must describe methodologically and ethically rigorous research with conclusions that are appropriately drawn based on the data presented.

Reviewer #1: Yes

Reviewer #2: Yes

Reviewer #3: Partly

Reviewer #4: No

3. Has the statistical analysis been performed appropriately and rigorously?

Reviewer #1: Yes

Reviewer #2: Yes

Reviewer #3: Yes

Reviewer #4: N/A

4. Have the authors made all data underlying the findings in their manuscript fully available (please refer to the Data Availability Statement at the start of the manuscript PDF file)?

Reviewer #1: Yes

Reviewer #2: Yes

Reviewer #3: Yes

Reviewer #4: Yes

5. Is the manuscript presented in an intelligible fashion and written in standard English?

Reviewer #1: No

Reviewer #2: Yes

Reviewer #3: No

Reviewer #4: Yes

6. Review Comments to the Author

Reviewer #1: Revised manuscript

In the title aren’t the household contacts the one with either positive or negative interferon gamma release assay

This tile may be misleading if the words “of individuals “

I suggest something like this

Detection of Plasma CXCL8 and MCP-1 as surrogate biomarkers of latent tuberculosis infection among house hold contacts in India

Under results

3.1 “cohort” characteristics

Please replace the word cohort with study

Under the same section , the grammar should be improves

For example the word sex remains as sex I plural not sexes

Under discussion “The study examined a fleet of cytokines” replace fleet with a panel

Reviewer #2: Thank you to the authors for addressing my comments. I still feel that plans for prospective clinical evaluation of the biomarkers identified in this study should be described in more detail, otherwise this manuscript lacks the clinical value it can potentially have.

Reviewer #3: I note this manuscript has been reviewed by 2 others before me, but I am a first time reviewer so have no response to comment 1.

Minor edits needed in figure 3A for clarity of data presentation - to be in line with the related text in section 3.3

The results and discussion are well presented, but the introduction and methodology need a number of grammatical changes

Reviewer #4: This study evaluated if a 17-plex cytokine assay is able to improve LTBI sensitivity.

The authors conclude that increased plasma CXCL8 and decreased MCP1, TNF, and IFNg are associated with LTBI and the authors conclude that CXCL8 and MCP1 could be surrogate biomarkers for LTBI. The major problem with the study is a lack of a proper reference standard. No tests of infection are perfect as evidence by the 5-25% of individuals with microbiologically confirmed TB who are negative by the TST, QFT, or Tspot. Host immunity to TB is initially proinflammatory/ cell-mediated, but eventually shifts towards anergy and therefore it is expected that any test of infection relying on the immune response will vacillate depending on the individuals host immunity. Host immunity to TB is extremely heterogeneous and therefore the sample size is inadequate. Therefore, the study groups (listed below) lacking a proper reference standard make the cytokine analysis very challenging to interpret.

The study consists of 76 volunteers (participants I think) including IGRA+ household contacts (n = 26) and IGRA- HHCs (n = 25) and 25 healthy controls (with no known TB exposure and IGRA-).

IGRA status was determined by only the QFT, which is probably acceptable since it is good concordance with the TST and Tspot.

Despite having the QFT (and therefore the left over supernatant), the study only evaluated the unstimulated plasma for cytokine levels. Unclear why this approach was taken?

A second major problem is that there is extremely large heterogeneity of host immunity among asymptomatic individuals due to HIV, smoking, nutrition, socioeconomic factors, helminth coinfections, etc. Therefore, considering the large expected variance in immune data, any test hoping to predict LTBI status would require much larger participants per group.

Minor:

How many individuals and how many replicates are unclear. The following lines are not understandable: “The study comprises 13 individuals in the x group (n=26).” Was the test performed twice for each individual? Why?

The results of Figure 2 are challenging to understand. What was the LOD of the assay for each cytokine? Why show the data in log scale? Is this log10 or log2?

7. PLOS authors have the option to publish the peer review history of their article (what does this mean?). If published, this will include your full peer review and any attached files.

**Do you want your identity to be public for this peer review?** For information about this choice, including consent withdrawal, please see our Privacy Policy.

Reviewer #1: **Yes: **ESTER LILIAN ACEN

Reviewer #2: No

Reviewer #3: **Yes: **Harriet Mayanja-Kizza

Reviewer #4: No
